# Heatwave Mortality in Summer 2020 in England: An Observational Study

**DOI:** 10.3390/ijerph19106123

**Published:** 2022-05-18

**Authors:** Ross Thompson, Owen Landeg, Ishani Kar-Purkayastha, Shakoor Hajat, Sari Kovats, Emer O’Connell

**Affiliations:** 1Extreme Events and Health Protection Team, UK Health Security Agency, London SW1P 3JR, UK; owen.landeg@nhs.net (O.L.); ishani.karpurkayastha@phe.gov.uk (I.K.-P.); emer.oconnell@phe.gov.uk (E.O.); 2Department of Social and Environmental Health Research, London School of Hygiene and Tropical Medicine, London WC1E 7HT, UK; shakoor.hajat@lshtm.ac.uk (S.H.); sari.kovats@lshtm.ac.uk (S.K.)

**Keywords:** heatwave, mortality, place of death, cardiovascular mortality, dementia

## Abstract

High ambient temperatures pose a significant risk to health. This study investigates the heatwave mortality in the summer of 2020 during the SARS-CoV-2 coronavirus (COVID-19) pandemic and related countermeasures. The heatwaves in 2020 caused more deaths than have been reported since the Heatwave Plan for England was introduced in 2004. The total and cause-specific mortality in 2020 was compared to previous heatwave events in England. The findings will help inform summer preparedness and planning in future years as society learns to live with COVID-19. Heatwave excess mortality in 2020 was similar to deaths occurring at home, in hospitals, and in care homes in the 65+ years group, and was comparable to the increases in previous years (2016–2018). The third heatwave in 2020 caused significant mortality in the younger age group (0–64) which has not been observed in previous years. Significant excess mortality was observed for cardiovascular disease, respiratory disease, and Alzheimer’s and Dementia across all three heatwaves in persons aged 65+ years. There was no evidence that the heatwaves affected the proportional increase of people dying at home and not seeking heat-related health care. The most significant spike in daily mortality in August 2020 was associated with a period of high night-time temperatures. The results provide additional evidence that contextual factors are important for managing heatwave risks, particularly the importance of overheating in dwellings. The findings also suggest more action is also needed to address the vulnerability in the community and in health care settings during the acute response phase of a heatwave.

## 1. Introduction

Episodes of high temperatures (heatwaves) pose a significant risk to health [1,2,3,4,5,6,7,8]. Global trends in the frequency, duration, and intensity of heatwaves show a steep increase since the 1950s [9]. Temperatures have increased in the UK and are higher than have been experienced previously. A new UK record of the max daily temperature of 38.7 °C was set during a brief but exceptional heatwave in July 2019. All the top 10 warmest years for the UK in the series from 1884 have occurred since 2002. Temperatures are projected to increase significantly, particularly in the scenarios with higher emissions [10]. The UK’s Third Climate Change Risk Assessment (CCRA3) has confirmed that hot weather remains an important risk to health for the UK population, and that current action, particularly to reduce overheating in dwellings, is insufficient to address both current risk and the increased future risk [11,12].

Individuals at risk of heat-related mortality include older people, those with chronic medical conditions, those with alcohol or drug dependence, or other mental health issues that affect behavior [13], and the homeless [14,15]. High indoor temperatures are an important determinant of heat risks, and older persons spend the majority of their time indoors [16,17,18].

The total estimated mortality associated with heatwaves in England in 2020 was 2556; the highest burden in any year since the introduction of the Heatwave Plan for England (HWP) in 2004 [19,20,21,22,23]. Summer (JJA) 2020 was 0.4 °C warmer than the long-term average (1981–2010) [24] but not comparable to the extremely hot summers of 2018 or 1976. June, July, and August 2020 all experienced hot-spells with the highest daily temperature of 37.8 °C recorded in England on 31 July [25]. 

2020 was an exceptional year with concurrent risks from the SARS-CoV-2 coronavirus (COVID-19) pandemic and significant countermeasures (including social distancing and lockdowns). Several individual risk factors are common across both hazards [26] such as older age and people with chronic disease. Further, everybody, including those high-risk individuals (to COVID-19) spent more time at home and potentially exposed to high indoor temperatures. Other factors that may also affect heat risks include: increased levels of social isolation [25]; changes to healthcare use [26,27]; and reduced capacity across care services to respond to a heatwave event [11,25].

This study aims to gain insight into how the impact of the 2020 heatwaves was different from other years, in terms of cause of death and place of death, and if these differences can be attributed to the COVID-19 pandemic and related countermeasures. The findings will help inform summer preparedness planning in future years as society learns to live with COVID-19.

## 2. Materials and Methods

### 2.1. Heatwave Definitions Used

Heatwave periods were identified using the definition outlined within published Heatwave Mortality Monitoring reports [19,20,21,22,23]. This is a day on which at least one region in England has been issued a Level 3 Heat-Health Alert (HHA) and/or Central England Temperature (CET) reached 20 °C and includes the days preceding and following the day on which the threshold was reached. Three heatwaves were recorded in 2020:Heatwave1 (H1) 23 to 27 June. East Midlands, West Midlands, East of England, London, Southeast and Southwest of England reached Level 3 HHA, and CET reached 20 °C;Heatwave2 (H2) 30 July to 1 August. No region reached a Level 3 HHA however CET reached 20 °C;Heatwave3 (H3) 5 to 15 August. East of England, London, and Southeast of England reached a Level 3 HHA, and CET reached 20 °C.

CET is a combined indicator of daily temperatures to represent central England (derived from four weather stations) [28]. Daily mean, maximum and minimum values were obtained from UK Met Office Hadley Centre for the years 2016, 2017, 2018, and 2020 (Figure 1).

### 2.2. All-Cause Mortality Data

Individual death records were obtained from the Office for National Statistics (ONS) for 01 June to 15 September 2020 (heatwave alert season). Mortality data included place of death, underlying cause of death, age, sex, and governmental region of usual residence. COVID-19 deaths, defined, during that period, as deaths reported either by ONS as COVID-19 on the death certificate or death for any reason within 60 days of a COVID-19 positive PCR test, were removed. A timeseries of daily all-cause excess deaths (excluding COVID-19) was plotted by place of death and cause of death by age.

### 2.3. Episode Analysis

Expected daily mortality was calculated as the mean value of the seven non-heatwave days before and after the period of interest (14 days in total) [23] with standard deviation (SD) standard error (SE) and 95% confidence intervals calculated assuming Poisson distribution. Where there were less than seven consecutive non-heatwave days, subsequent non-heatwave days were used to ensure a total of 14 non-heatwave days were used in calculating the baseline. Heatwave attributable deaths were calculated as observed deaths in the heatwave period minus the expected number of deaths. Daily z-scores indicating the number of standard deviations the observed mortality count was above or below the expected daily value were calculated for each episode with a z-score of 3 (3SD) considered to be significant [23]. The 95% confidence intervals for the estimated daily excess mortality values were also calculated based on the estimated baseline values. 

### 2.4. Heatwave Impacts by Age, Cause of Death, Place of Death and Region

Table 1 describes the variables used in subgroup analyses based on previous epidemiological studies of heat effects in England. Categories of the place of death were based on PHE’s National End of Life Care Intelligence Network (NEoLCIN) guidance [29]. Only regions recording significant all-cause excess mortality were included in each episode analysis.

Furthermore, an analysis of heatwave deaths by place of death in 2020 was compared to previous analyses from heatwaves between 2016 and 2018 to investigate possible changes in healthcare use and potential heightened vulnerabilities during the pandemic.

## 3. Results

### 3.1. Descriptive Analysis

The mean daily CET series during the summer season of 2020 was not distinctly different from the mean daily CET series for 2016, 2017, and 2018 over the same period. This can be observed in Figure 1. However, the August period does stand out in that temperatures were high, but for a prolonged period.

Peaks in daily all-cause deaths exceeding three standard deviations (3SD) significance threshold were observed during each heatwave period for the 65+ age group, with a total estimate of the all-cause excess mortality of 1807 (95% CI 1575 to 2037). Estimates per heatwave episode are displayed in Figure 2. For the <65 years group, peaks in the daily all-cause deaths only exceeded the 3SD threshold during the third heatwave with a total estimate of the all-cause excess mortality of 207 (95% CI 112 to 301). There was no obvious trend observed immediately following the daily peaks suggestive of a short-term mortality displacement for either age group.

Across both age groups and all settings, over 60% of the total heatwave associated all-cause excess mortality occurred during H3 with two distinct phases corresponding to the temperatures observed. Deaths increased as temperatures increased at the start of H3, falling to expected levels as temperatures leveled off. The subsequent surge in temperatures from 10th August represented a second phase with even larger peaks in daily deaths (Figure 2).

### 3.2. Place of Death

Significant (>3SD) excess all-cause mortality among the 65+ years group was observed during each heatwave for deaths occurring at home and in care homes (Table 2). Significant excess all-cause mortality in the 65+ years group was also observed during the first and last heatwave for deaths occurring in hospitals. This differs from heatwave deaths between 2016 and 2018 where a significant excess of deaths in hospitals was observed during the first heatwave of the season only. During H3, excess all-cause mortality was observed across all places of death, with the exception of deaths occurring “elsewhere”. There were no obvious post-heatwave deficits in all-cause mortality for any place of death following the peaks identified, suggesting no significant short-term mortality displacement. 

Among those in the <65 years group, significant (>3SD) excess all-cause mortality was observed for deaths occurring at home, in care homes, and in hospitals during H3. This differs from previous heatwaves from 2016 to 2018 where no significant excess was observed in this age group. 

The increase in deaths occurring at home, in care homes, and in hospitals, overall, during all three heatwave events were comparable to the proportional increase in daily mortality by place of death observed in 2016, 2017, and 2018. So, whilst the absolute number of excess deaths associated with the heatwave in 2020 was high (2556) [22], Figure 3 indicates that the relative distribution of these by place of death was broadly consistent with previous years. 

### 3.3. Cause of Death

In the 65+ years group, deaths from circulatory, respiratory disease, Alzheimer’s and Dementia were increased during all three heatwaves (Table 3). There were no obvious deficits in mortality following each daily peak suggesting minimal mortality displacement. Significant excesses in cancer deaths were observed during the first two heatwaves but not the third. There were no other consistent patterns in other causes of death. 

Interpretation of the cause of death for excess mortality in the <65 years groups is challenging due to the small number of deaths in each category.

## 4. Discussion

### 4.1. Summary of Findings

The impact of the heatwave in 2020 was significantly larger than for previous events. The pattern of mortality (in 65+) age group and by cause of death is consistent with previous events in England [4,19,20,21,22,30,31]. The impact is greater than would be expected from the temperature exposures and therefore the pandemic and lockdown likely had an important effect on heatwave mortality. 

The proportional increase in deaths occurring in each place of death category in 2020 was comparable to previous years, with a 15% to 24% increase in mortality occurring at home; 19% to 30% increase in deaths occurring in care homes; deaths in hospital 6% to 26% increase in mortality occurring in hospitals across all years. As indicated in Figure 3, 2020 estimates are not the upper end of this range. The main causes of death which observed a significant increase in 2020 heatwaves included cardiovascular disease, respiratory disease, and Alzheimer’s and Dementia, which is aligned with the published literature [32]. Thus, there was no evidence that the 2020 heatwaves increased the proportion of people dying in different settings, or that the pandemic shifted the main underlying causes of death during a heatwave. 

### 4.2. Potential Contributing Factors

An important impact in 2020 was the excess mortality in adults (<65 years group) during the third heatwave. Reasons behind the large excess within this age group may also be related to increased indoor exposure in vulnerable or institutionalized adults. Contributing factors include increased population risk due to the sequelae of COVID-19 infection. Local lockdowns are unlikely to have played a part in affecting risks in the general adult population as only Greater Manchester and Leicester were under local restrictions at the time. Health surveillance systems which had highlighted a large fall in help seeking behavior in Emergency Department attendances in March 2020 [27] showed a return to near 2019 levels by August 2020.

Recent evidence suggests that the observed excess all-cause mortality in 2020 was significantly higher than would have been expected based on the temperature mortality relationship alone (5556 (95% CI 2139 to 2929) observed, 1783 (95% CI 1643 to 2981 expected) [33]. This along with previous evidence that there is significant temporal variation in the temperature mortality relationship across years [34] suggests that the impacts which are observed year on year are likely to also be influenced by other contextual factors. In 2020, the pandemic clearly played a role and may have amplified the observed impacts, however, it is not possible to determine the precise mechanisms with the mortality data currently available.

As new generations of Heat-Health Warning systems are developed across the world, they need to consider the interaction of heatwaves with other shocks or hazards, as well as the contextual factors which can increase risks during these events.

The third heatwave was uniquely long (11 days) and biphasic with an initial temperature increase followed by a surge about halfway through the episode. The time series analysis across both age groups, and by all settings and causes of deaths indicates that the peak of the impacts occurred during the second phase of H3. This was a period during which there were several tropical nights, where overnight temperatures remained above 20 °C, a known risk factor during heatwaves [35]. The prolonged nature and high night temperatures may have been an important factor in relation to mortality, specifically amongst the <65 years group as other studies have reported high overnight temperatures as a key factor for younger groups [35].

There were no obvious mortality deficits observed in the days following the high temperatures. Short-term mortality displacement due to high temperatures is possible for some of the mortality impacts, since any deficits may be distributed over many months following the initial excess. Our analysis suggests that the majority of the excess mortality observed was not due to deaths moved forward in the short term. 

### 4.3. Limitations of This Study

There are several limitations in this analysis that need to be acknowledged. First, there are a number of methodological limitations around the calculation of the baseline. The method used in estimating baseline mortality in 2020 differed from previous years due to anticipated distortion to the baseline by COVID-19 deaths. However, the precise approach for dealing with the effect of COVID-19 deaths in assessing excess mortality is an area in general that still requires attention as society moves into a post-pandemic world and the full effect (direct and indirect) is understood. Additionally, the duration of H1 and H2 were less than the comparison period (14 days), therefore there may have been a day-of-week effect. However, it was deemed that any variation by day of the week would likely be small. 

Secondly, COVID-19 deaths were extremely low during each heatwave and in summer 2020 in general, and by excluding these from this analysis, it was possible to assess the impact on heatwave mortality during the COVID-19 pandemic, independent of the infection itself. Conversely, however, it was not possible to assess the potential impact of heat exposure on individuals with COVID-19. This may be an area in which future research may want to focus to gain further insight into how future impacts during heatwaves may manifest as the infection becomes endemic. 

Finally, our study does not attempt to disentangle other possible behavioral or environmental risk factors for the large increase in excess deaths observed in 2020. For example, Syndromic Surveillance Systems demonstrated that there was a significant drop in the numbers of people attending health care services during the pandemic. Weekly excess mortality reports from 2020 also show that overall, there was a shift to more deaths occurring at home and a reduction of deaths occurring in hospitals as people avoided risky locations. Furthermore, the population may have been spending more time outside, with parks and public outside spaces frequently much busier than normal. This shift to spending more time outside and potentially in direct sunlight may have also contributed to increased risk. Further work is required to investigate the complex relationships that may have contributed to the mortality observed and the intersecting risk of heatwaves and the COVID-19 pandemic.

## 5. Conclusions

This is the first study to explore heatwave mortality patterns during the COVID-19 pandemic. In addition, this is the first analysis of the place of death in England during heatwave episodes since the 2003 heatwave which swept Europe, killing an estimated 40,000 people [36], with over 2000 additional deaths in England and Wales alone [30]. However, we were able to contextualize the 2020 analysis by comparing the data to observations from recent years. These data indicate that while deaths occurring at home, in care homes, and in hospitals did indeed increase during heatwave episodes in 2020, the percentage increase above the baseline was comparable to previous years. This should however be viewed in the context that during the pandemic, evidence suggests that there has been a shift of where deaths have occurred, with a displacement of deaths occurring in hospitals to homes previously reported [37,38].

Findings from this study shed light on where excess mortality is occurring in England and can be used to shape heat-health communication strategies to ensure that those with a role in reducing risk for vulnerable individuals are aware of the risks and of what actions can be taken. For example, a consistent trend of excess all-cause mortality during heatwaves in care homes and in hospitals were observed, highlighting the need for mitigating actions to reduce those harms; for example, increased frequency of hydration checks or implementation of heatwave response plans that include cooling interventions. These findings indicate that vulnerability to heat is not equally distributed across hospital patients or care home residents, raising the potential of a risk profiling tool for these settings. 

Action is also needed in the community to address deaths that occur at home. Certain types of housing are prone to high indoor temperatures [39,40] and therefore increase the risk of exposure. Future changes to the delivery of care, with a trend of increasing and complex care at home, an aging population, and the increased probability of more frequent, longer, and more intense episodes of heat also require health and social care services to develop plans to protect individuals at home during the acute phase of a heatwave event as well as longer-term strategies to reducing population level risks now and in the future. 

As the climate continues to warm and episodes of extreme heat become more frequent, intense, and prolonged, additional capacity and resources across the health and social care system will be required. Despite the current focus on preparedness for future pandemic threats, there remains an urgency to ensure that we are also prepared for our future climate [9].

## Figures and Tables

**Figure 1 ijerph-19-06123-f001:**
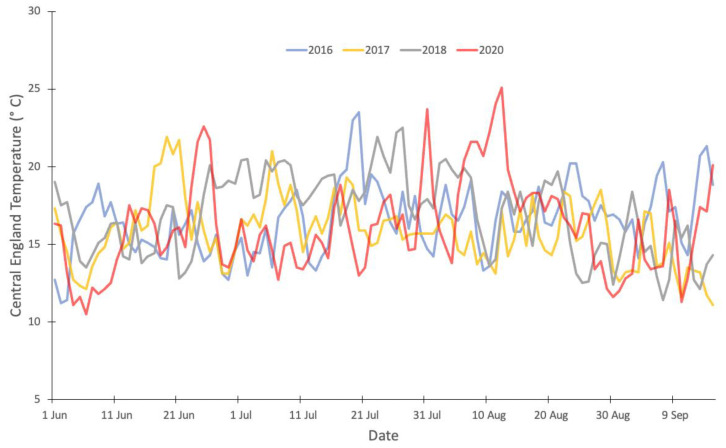
Mean Daily Central England Temperature series for 2016 (blue), 2017 (yellow), 2018 (grey), and 2020 (red).

**Figure 2 ijerph-19-06123-f002:**
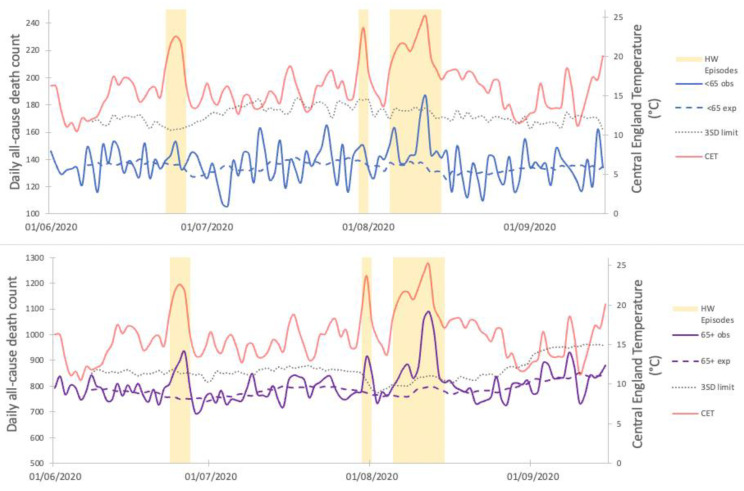
Three mortality peaks associated with the heatwave events identified for both the <65 years group (top panel) and the 65+ years group (bottom panel). Both panels also display the calculated daily expected deaths (dashed lines) and the 3 standard deviation threshold used to determine significance (dotted line). Heatwave episodes are highlighted on both panels with CET included for context.

**Figure 3 ijerph-19-06123-f003:**
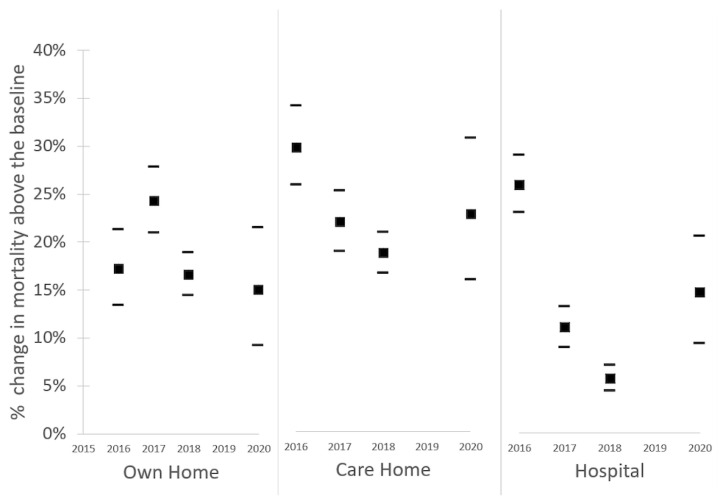
Black squares represent the percentage change in mortality above the baseline occurring at home, in care homes, and in hospitals during heatwave days compared to non-heatwave days for the years 2016, 2017, 2018, and 2020. Black dashes represent the calculated 95% confidence interval.

**Table 1 ijerph-19-06123-t001:** Variables used in analysis with the total number of deaths by category and percentage of total number of deaths over the study period, 1 June to 15 September 2020. ICD 10 codes are provided for each cause of death category.

Variable	Category	Number of Deaths	% Total
Age	0–64	17,675	15%
	65+	102,102	85%
Sex	Male	59,482	50%
	Female	60,295	50%
Region	Northeast England (NE)	6592	6%
	Northwest England (NW)	16,865	14%
	Yorkshire and Humber (Y&H)	12,367	10%
	East Midlands (EM)	11,040	9%
	West Midlands (WM)	13,624	11%
	East of England (EoE)	13,695	11%
	London (LON)	11,456	10%
	Southeast England (SE)	20,027	17%
	Southwest England (SW)	14,111	12%
Place of death categories defined by NEoLCIN Underlying cause of death	Own home	40,772	34%
	Care home	25,670	21%
	Hospital	6924	6%
	Hospice	46,097	38%
	Elsewhere *	354	<1%
Underlying Cause of death (ICD10 codes)	Ischaemic heart disease (I20 to I25)	948	2%
	Stroke—*cerebrovascular diseases* (I60 to I69)	6891	6%
	Other circulatory diseases (I00 to I19, I26 to I59 and I70 to I99)	30,955	26%
	Cancer (C00 to C97)	4086	3%
	Acute respiratory infections—*including flu/pneumonia* (J10 to J11, J12 to J18)	3410	3%
	Chronic lower respiratory diseases—*including asthma* (J40 to J47)	5190	4%
	Other respiratory diseases (J00 to J09, J19 to J39, J60 to J99)	8415	7%
	Dementia and Alzheimer’s disease (F01,F03,G30)	13,843	12%
	Diseases of the urinary system (N00 to N39)	1843	2%
	Cirrhosis and other liver diseases (K70, K73 to K74)	2016	2%
	Parkinson’s disease (G20)	1660	1%
	Suicide (X60 to X84)	991	1%
	Accidents (V01 to X59)	1948	2%

* The “elsewhere” category for the place of death analysis is defined as deaths not occurring at the other categories. For example, psychiatric hospitals, schools, convents and monasteries, university, and college halls of residence, young offender institutions, secure training centers, detention centers, prisons, and remand homes.

**Table 2 ijerph-19-06123-t002:** Estimated excess all-cause mortality by age group and place of death in affected regions during heatwaves in England, summer 2020. Bold values indicate all-cause excess mortality estimates considered to be statistically significant (above 3SD), with values within the brackets the 95% Confidence intervals. Heatwave1 (H1) occurred 23 to 27 June affecting EM, WM, EoE, LON, SE, and SW; Heatwave2 (H2) occurred 30 July to 1 August. All regions affected; Heatwave3 (H3) occurred 5 to 15 August affecting EoE, LON, and SE.

Place of Death Category	Age Group	Heatwave Event		
		H1	H2	H3
**Deaths at home**	**<65**	5 (−14 to 24)	0 (−8 to 8)	**61 (3 to 119)**
	**65+**	**90 (50 to 130)**	**20 (2 to 39)**	**416 (289 to 542)**
**Deaths in care homes**	**<65**	−2 (−6 to 3)	N/A	**28 (12 to 43)**
	**65+**	**87 (53 to 121)**	**54 (39 to 70)**	**463 (353 to 573)**
**Deaths in hospitals**	**<65**	15 (−4 to 33)	6 (−3 to 15)	**63 (0 to 126)**
	**65+**	**146 (105 to 187)**	19 (−1 to 39)	**437 (298 to 576)**
**Deaths in hospices**	**<65**	4 (−4 to 12)	4 (0 to 8)	14 (−14 to 43)
	**65+**	0 (−17 to 17)	4 (−3 to 11)	**64 (14 to 115)**
**Deaths in all settings**	**<65**	**23 (3 to 23)**	9 (0 to 18)	**175 (110 to 240)**
	**65+**	**322 (274 to 371)**	**100 (78 to 123)**	**1384 (1225 to 1543)**

**Table 3 ijerph-19-06123-t003:** Estimated excess mortality for specific underlying causes of death by age group in England during three heatwaves in 2020. Bold values indicate all-cause excess mortality estimates considered to be statistically significant (above 3SD), with values within the brackets the 95% Confidence intervals. Heatwave1 (H1) occurred from 23 to 27 June affecting EM, WM, EoE, LON, SE, and SW; Heatwave2 (H2) occurred from 30 July to 1 August. All regions affected; Heatwave3 (H3) occurred from 5 to 15 August affecting EoE, LON, and SE.

Cause of Death	Age Group	H1	H2	H3
Ischemic HD (ICD10 code: I20 to I25)	<65	−4 (−7 to −2)	−9 (−11 to −7)	**12 (9 to 15)**
	65+	**18 (14 to 22)**	**9 (6 to 13)**	7 (0 to 13)
Cerebrovascular diseases (ICD10 code: I60 to I69)	<65	**11 (8 to 13)**	**11 (8 to 13)**	**9 (5 to 13)**
	65+	**25 (17 to 34)**	0 (−7 to 7)	**69 (56 to 83)**
All circulatory deaths (ICD10 code: I00 to I99)	<65	**29 (22 to 35)**	**6 (1 to 12)**	**33 (22 to 44)**
	65+	**115 (97 to 133)**	**45 (31 to 60)**	**408 (380 to 435)**
Acute respiratory infections (including flu/pneumonia) ICD10 code: J10 to J11, J12 to J18)	<65	1 (−1 to 3)	−3 (−4 to −1)	−4 (−7 to −1)
	65+	**19 (13 to 25)**	**15 (10 to 20)**	**16 (6 to 25)**
Chronic lower respiratory deaths (ICD10 code: J40 to J47)	<65	−3 (−6 to 0)	−3 (−5 to −1)	9 (5 to 13)
	65+	**30 (23 to 38)**	**14 (8 to 19)**	**118 (107 to129)**
Other respiratory deaths (ICD10 code: J00 to J09, J19 to J39, J60 to J99)	<65	−11 (−15 to −8)	3 (−5 to 0)	9 (4 to 14)
	65+	**72 (63 to 82)**	**11 (4 to 19)**	**192 (187 to 206)**
Alzheimer’s and Dementia (ICD10 code: F01,F03,G30)	<65	N/A	N/A	0 (−1 to 2)
	65+	**47 (34 to 60)**	**23 (13 to 33)**	**62 (42 to 83)**
Parkinson’s disease (ICD10 code: G20)	<65	N/A	N/A	N/A
	65+	3 (−1 to 8)	**8 (4 to 11)**	−1 (−8 to 7)
All cancer deaths (ICD10 code: C00 to C97)	<65	**8 (5 to 11)**	**4 (1 to 6)**	−10 (−15 to −5)
	65+	**17 (11 to 24)**	**8 (3 to 13)**	−10 (−20 to 0)
Diseases of the urinary system (ICD10 code: N00 to N39)	<65	N/A	N/A	**5 (4 to 7)**
	65+	−13 (−18 to 8)	−5 (−8 to −1)	**9 (2 to 16)**
Cirrhosis and other liver diseases (ICD10 code: K70, K73 to K74)	<65	−5 (−9 to −1)	−9 (−12 to −5)	−22 (−28 to−15)
	65+	3 (0 to 6)	**4 (2 to 6)**	**14 (10 to 18)**
Suicide (ICD10 code: X60 to X84)	<65	**5 (3 to 7)**	−2 (−4 to 0)	−14 (−18 to 11)
	65+	−2 (−4 to 1)	**6 (4 to 8)**	4 (0 to 8)
Accidental death (ICD10 codes: V01 to X59)	<65	−4 (−7 to −2)	−9 (−11 to −7)	**12 (9 to 15)**
	65+	**18 (14 to 22)**	**9 (6 to 13)**	7 (0 to 13)

## Data Availability

Not applicable.

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
