# Peer review of "Heatwave Mortality in Summer 2020 in England: An Observational Study"

_ijerph, 2022, doi:10.3390/ijerph19106123_

Round 1
Reviewer 1 Report
This manuscript compares the heatwave-attributable mortality in 2016-2018 and 2020 by means of statistical analysis. The analysis showed that heatwave-attributable mortality in 2020 was significantly higher than in 2016-2018. Still, the proportion of deaths occurring in each place of death category in 2020 was comparable to previous years. The manuscript also analyzed the possible reasons why the heatwave-attributable mortality in 2020 was different from previous ones under the dual threats of the pandemic and heatwave.
Overall, this is a very interesting and meaningful study. This study also proposed possible mitigation measures to reduce mortality during heatwaves. I do have several suggestions for substantive improvement of the manuscript, followed by more suggestions/comments, outlined below.
Major suggestions:
2020 was mentioned in the manuscript's abstract as a year in which both the pandemic and relative countermeasures may pose risks. But I regret that what is laid out in the abstract does not mention the impact of the pandemic on heatwave-attributable mortality. Also, I don't want to see in the abstract the statement that " the main causes of death aligned with the published literature." This will result in readers not being able to obtain critical information about the manuscript research in the first place.
Although the introduction describes the related research on heatwave-attributable mortality and the related research on heatwave-attributable mortality caused by the pandemic, at the end of the opening, it does not clearly address the scientific question to be studied in the manuscript. I suggest that the critical point of the introduction is not only to summarize the previous relevant research but, more importantly, to propose the key questions to be solved by this research. The problems that need to be solved at the end of the introduction should correspond to the results and discussion of the manuscript. And the results and discussion are to solve the scientific problems raised in the introduction.
I have no objection to the fact that the current Discussion section does not have subheadings. However, in order to make the logic of the discussion part of the manuscript clearer, I suggest classifying it in the discussion. The Discussion section should be categorized according to the issues that need to be explored in the manuscript. For example: 4. Discussion 4.1 Changes and characteristics of heatwave-attributable mortality in 2020. 4.2 Pandemic's potential impact on heatwave-attributable mortality. 4.3 Suggestions / strategies for mitigating heatwave-attributable mortality…
Minor suggestions:
Ln 19-20: What are the conclusions of the published literature? What is the reason for this phenomenon (the proportion of deaths occurring in each place of death category in 2020 was comparable to previous years)?
For 3 sections: It is proposed to add a graph comparing the temperature of 2016-2018 with 2020 during the heat wave.
Ln 51-53: It is suggested that the scientific problems to be solved in this study should be elaborated in more detail.
Ln182-184: It is recommended to add more evidence to illustrate this conclusion.
Ln 197-206: “The observed excess all-cause mortality in 2020 was significantly higher than would have been expected based on the temperature mortality relationship alone.” Could this prove that the pandemic played an important role in heatwave-attributable mortality?
Some analyses in the study are qualitative, and there are still some hypothetical analyses. It is suggested to further explain the limitations of the study.
Author Response
Thank you for these valuable responses to our manuscript. Please see below responses to all review comments.

Reviewer 2 Report
A review on the manuscript in International Journal of Environmental Research and Public Health entitled „Heatwave mortality in summer 2020 in England: an observational study“.
The article examines the mortality due to heatwaves observed in 2020, comparing the mortality observed in 2020 with the previous heatwaves in 2016–2018.
Broad comments
The description of the statistical methods used is not sufficient. Therefore, there is no data on whether the prerequisites for using these methods are met. For example, frequency distributions of data have not been analyzed. For example, on line 85 is written "Assuming a Poisson distribution...", but there is no information as to which method was used to determine whether the data correspond to the Poisson distribution. Computing a z-score (line 84) requires knowledge of the mean and standard deviation of the complete population to which a data point belongs; if one only has a sample of observations from the population, then the analogous computation using the sample mean and sample standard deviation yields the t-statistic. If the data is not distributed normally, standard deviation (lines 107, 110, 127, 137), t-statistic, etc. will not have real content. The assumptions for the use of statistical methods and the response of the data to these assumptions and constraints need to be presented in more detail.
The article needs technical revision.
Specific comments
Academic writing should be objective. If it is subjective or emotional, it will lose persuasiveness and may be regarded as relying on emotion rather than building a reasonable argument based on evidence. The language or informal writing should therefore be impersonal, and should not include personal pronouns. For most subject areas the writing is expected to be objective. For this the first person (I, we, me, my, etc.) should be avoided. In this article on line 94 is written “we compared …", on line 230 is writtwen „we are able „, on line 232 is written„we have not been“, on line 244 is written „we were able“, on line 252 is written „We found“, on line 270 is written „we also prepared“. Eliminating personal pronouns from writing is highly recommend.
It must always be a space between the numerical value and unit symbol except the plane angle and percent – line 38 „0.4°C“ must be „0.4 °C“, etc, etc; lines 40, 59, 63, 65, 67 and 213.
There is no space between the words on line 26 "health[1-8]", on line 270 „climate[9].
There is no dot at the end of the sentence (line 215).
References to sources must be in the same style in the article, for example, line 33 is [12,13], but lines 177 and 260 also use spaces - [4, 17-20, 27, 28] and [36, 37].
When referring to tables, the reference must begin with an uppercase letter, but line 128 contains "table 2".
Lines 48 and 49 use the incomprehensible or unexplained term "levels of social isolation24".
Author Response
Thank you for these valuable comments. Please see the table attached for specific responses to the reviewers comments.

Reviewer 3 Report
Review for Heatwave mortality in summer 2020 in England: an observational study, by Thompson et al. (2022).
This manuscript attempts to examine the heatwave mortality in summer 2020 in England. Overall, the study is interesting using well monitoring and updated scenario across England, however, there is some major flaws and questions from reviewer before going to consider for publication.
-The abstract is very rough; the objectives seem not clear. The author needs to revise the whole abstract. I think the main findings is lacking in the abstract and should be revised based on the results. Besides, the abstract need the concluding remarks for the community beneficial and role of mitigation strategies across region.
In introduction, please discuss the recent global warming in term of extreme temperatures across other parts of the world and correlate their impacts scenarios with your own region.
However, there is lack of driver’s propagation and their onsets. It should be discussed with proper recent literature.
In the discussion section, extend the paragraph in terms of Heatwaves, droughts, hydrological process, and their impacts on other parts of the globe.
The conclusion section also needs attention and concluding remarks, to provide some remarks regarding what has been found from the results and their contribution to the community mitigation processes in the future.
Author Response
Thank you for these valuable comments. Please see the attached table for responses to all comments.

Reviewer 4 Report
The authors have compactly described the outline of excess deaths in the three heat waves in England in 2020 summer. The article is basically acceptable. However, the comparison between 2020 and previous years should be based on more quantitative evidence.
[Main comments]
(1) One of the subjects of the paper is the difference between 2020 and previous years. The discussion should be made as quantitatively as possible by presenting data.
The authors have written:
@ "the absolute number of excess deaths associated with heatwave in 2020 was high" (Line 143)
@ "The impact of the heatwave in 2020 were significantly larger than for previous events." (Line 175)
@ "the impact is greater than would be expected from the temperature exposures" (Line 178)
However, excess mortality during heat waves will depend on the magnitude and temporal variation of temperature, which should vary from year to year. A higher temperature peak will result in a greater number of deaths. Can you say that there were more deaths in 2020 than in 2016, 2018 even taking those factors into account? The evidence for this should be presented as much as possible.
(2) Some additional explanations are desired for following terms.
@ Central England Temperature (CET) (Line 58) --- Is it defined by daily average temperature? In what way?
@ Daily z-scores (Line 84)
@ standard deviations (Line 107) --- How was it calculated? For what period?
[Other comments]
@ In Table 1, "Other circulatory diseases (00 to I19" --- I00 to I19
@ Please adjust the position of "Category" and "Number of deaths" in Table 1 so that they are printed on the same line.
@ Please add tick marks to the x-axis of Table 1.
@ Please explain the meaning of bold letters and figures in parentheses in Table 2.
Author Response
Thank you for these valuable comments. Please see attached for responses to all comments.

Round 2
Reviewer 2 Report
A review on the manuscript in International Journal of Environmental Research and Public Health entitled „Heatwave mortality in summer 2020 in England: an observational study“.
The article examines the mortality due to heatwaves observed in 2020, comparing the mortality observed in 2020 with the previous heatwaves in 2016–2018.
Broad comments
The authors of the article have made a number of improvements and additions that have improved the quality of the article.
Research methods have been described at satisfactory level. The conclusions are based on analysis and are adequate.
The article needs minor technical revision.
Specific comments
Line 35 is written "risk. [10,11].". One point is superfluous. If the reference is after the end point of the sentence, the reference is valid to the sentence, if the reference is after the end point of the last sentence of the paragraph, the reference is to the whole paragraph. No reference is made between the two points.
Line 76 is written "stations][27]." There is no space in front of the reference.
Line 195 has no end point.
Heading 265 does not begin with a capital letter.
The formatting of the text in the whole article must be in the same style. But paragraphs from line 179 to 334 are not formatted using "Paragraph / Indentation / Special / First line" as in the previous text.
Author Response
Thank you for these useful technical comments – these points have now all been addressed within the manuscript.
Reviewer 3 Report
Please find in attachments

Author Response
Thank you for your comments. Please see attached our responses to specific comments.

Reviewer 4 Report
I appreciate the authors' effort of revision. The paper is acceptable after minor correction.
@ Line 10 "summer of 2002" --- 2020
@ Line 216 "As indicated in figure 4" --- Fig.3?
@ Please tick marks on the x-axes of Figs.1 and 2, so that we can read the exact dates in the graph. Please do the same to the y-axes of Figs.1-3, except that on the right of Fig.2.
Author Response
Thank you for these comments, all points including ticks on the charts have now been addressed.